# Neuromuscular Adaptations after an Altitude Training Camp in Elite Judo Athletes

**DOI:** 10.3390/ijerph18136777

**Published:** 2021-06-24

**Authors:** Katja Tomazin, Filipa Almeida, Igor Stirn, Paulino Padial, Juan Bonitch-Góngora, Antonio J. Morales-Artacho, Vojko Strojnik, Belen Feriche

**Affiliations:** 1Faculty of Sport, University of Ljubljana, 1000 Ljubljana, Slovenia; igor.stirn@fsp.uni-lj.si (I.S.); vojko.strojnik@fsp.uni-lj.si (V.S.); 2Department of Physical Education and Sport, Faculty of Sport Sciences, University of Granada, 18071 Granada, Spain; lua_d_prata@hotmail.com (F.A.); ppadial@ugr.es (P.P.); juanbonitch@ugr.es (J.B.-G.); antonio.morales@insep.fr (A.J.M.-A.); mbelen@ugr.es (B.F.); 3Laboratory Sport, Expertise and Performance (EA 7370), Research Department, French Institute of Sport (INSEP), 75012 Paris, France

**Keywords:** elite athlete, strength training, twitch, H-reflex, voluntary activation

## Abstract

The aim of this study was to investigate neuromuscular adaptations in elite judo athletes after three weeks of power-oriented strength training at terrestrial altitude (2320 m). Nineteen men were assigned to altitude training (AL) (22.1 ± 2.3 years) and sea level training (SL) (22.6 ± 4.1 years). Neuromuscular assessment consisted of: (1) maximal isometric knee extensor (KE) torque, (2) KE rate of torque development (RTD), (3) quadriceps activity and voluntary activation, (4) soleus H-reflex, (5) quadriceps single (T_TW_) and double twitch torque (T_DB100_) and contraction time (CT_TW_). There were no significant differences between groups at baseline for any of the observed parameters. Significant differences were found between groups in terms of change in RTD (*p* = 0.04). Cohen’s d showed a positive significant effect (0.43) in the SL group and a negative significant effect (−0.58) in the AL group. The difference between groups in changes in CT_TW_ as a function of altitude was on the edge of significance (*p* = 0.077). CT_TW_ increased by 8.1 ± 9.0% in the AL group (*p* = 0.036) and remained statistically unchanged in the SL group. Only the AL group showed a relationship between changes in T_TW_ and T_DB100_ and changes in RTD at posttest (*p* = 0.022 and *p* = 0.016, respectively). Altitude induced differences in muscular adaptations likely due to greater peripheral fatigue.

## 1. Introduction

Judo is a complex, high-intensity sport where success in competition requires a wide range of skills and good technical knowledge [1,2,3]. A judo match can last from a few seconds to eight minutes, but typically lasts three minutes, with 20–30 s of activity periods and 5–10 s of interruption [2]. Effective judo techniques require highly developed maximal strength and power, performed under adverse metabolic conditions [4,5]. In order to meet these specific anaerobic and aerobic metabolic demands of judo combat, it is of primary importance that elite judo athletes optimize their conditioning training, focusing on the development of power along with endurance.

Altitude training (for 2–4 weeks) is a common strategy of elite athletes that allow to benefit for explosive movements and aerobic and anaerobic metabolism [6]. In general, elite athletes use various training strategies, the most common being the traditional prolonged exposure to natural or artificial altitude, which has become known as “live high-train high”, where athletes live and train at a terrestrial altitude (i.e., 1800–3300 m a.s.l.) [7]. While the efficacy of various altitude training strategies on aerobic abilities due to a hypoxic erythropoietic effect is widely recognized [8], the optimization of anaerobic abilities, such as maximal or explosive strength, is still heavily debated. Currently, there is a large gap in the understanding of acute and chronic neuromuscular adaptation due to strength training in terrestrial and artificially induced hypoxia. Nonetheless, some studies have already shown a positive effect on maximal and explosive strength when athletes were acutely exposed to higher altitudes. For example, acute exposure to moderate altitude (i.e., 2320 m) results in increases in one-repetition maximum (RM), movement velocity, and power during bench press execution [9]. In addition, the only study conducted with judokas showed that acute exposure to moderate altitude (i.e., 2320 m) increased peak velocity during countermovement jumps loaded with 25% and 100% of body weight [10]. However, none of the above studies examined the neural factors that might contribute to the acute increase in explosive power. Similarly, chronic exposure to moderate altitude during strength training also shows increases in squat jump [11] and countermovement jump performance [12], although the exact neuromuscular outcomes underlying these improvements also remain largely unknown. Furthermore, power-oriented strength training in artificially induced hypoxia results in beneficial musculoskeletal changes and increases strength and muscular endurance [13].

An increase in maximal isometric torque after short-term strength training period (i.e., after 3–6 weeks) can be expected due to the increase in neural drive [14]. The increase in neural drive is commonly analyzed by estimating voluntary activation levels, i.e., using twitch interpolation techniques [15]. Although not consistently demonstrated in all studies [16,17], an increase in neural drive has already been reported during maximal isometric contraction following strength training in artificially induced hypoxia [18]. However, caution should be exercised when transferring findings from artificially induced hypoxia to the application of terrestrial hypoxia and to elite judo athletes. Therefore, it remains an open question whether short duration explosive strength training combined with endurance training performed at terrestrial altitude would induce a different neural adaptation than similar training performed at sea level. A more detailed look at neural adaptation following short duration strength training is commonly assessed by changes in the H-reflex [19]. The H-reflex is an estimate of the excitability of the alpha motoneuron when presynaptic inhibition, intrinsic excitability of the soma and axon of the alpha motoneuron, the strength of Renshaw inhibition, and the excitability of the sarcolemma remain constant [20,21,22]. It should be emphasized here that an increase in spinal excitability has already been demonstrated with acute exposure to artificially induced [23] and terrestrial hypoxia [24]. However, the chronic effect of short-term strength training at terrestrial altitudes (2000–3000 m a.s.l.) on spinal excitability remains to be elucidated. To summarize the above results, the application of strength training at altitude might induce greater neural adaptation compared to similar training at sea level.

Increases in maximal isomeric torque from strength training are also commonly attributed to muscular factors such as skeletal muscle hypertrophy and architectural changes [14]. It has been shown that increased metabolic load during strength exercise in hypoxia can lead to the recruitment of higher threshold motor units [25], resulting in a greater accumulation of metabolic by-products [26] and hypertrophic signals [27], which can increase the cross-sectional area of muscle fibers in less time than similar training at sea level [28]. A detailed investigation of twitch contractile properties could provide further insight into muscular adaptation to resistance training at terrestrial altitude in elite judo athletes. To our knowledge, there is only one study that has examined twitch contractility adaptation following short-term simultaneous strength and endurance training in elite swimmers at terrestrial altitude. Specifically, Tomazin and colleagues [24] showed that strength and endurance training at altitude leads to an increase in twitch contractile amplitude, likely due to the upregulation of physiological processes beyond sarcolemma, i.e., mechanisms of excitation-contraction coupling, which in turn would increase Ca^2+^ available for cross-bridge formation, allowing for stronger contractions [29,30]. Currently, there is no information on how short-term power-oriented strength training at moderate altitude affects the contractile properties of twitch in elite athletes.

Therefore, the aim of this study is to investigate whether there is an improvement in neuromuscular adaptations in elite judo athletes after short-term power-oriented strength training at moderate altitude (2320 m) performed in at the end of a special preparation period. We hypothesized that short-term power-oriented strength training at moderate altitude might increase neural and muscular adaptations compared to similar training at sea level. Therefore, a greater increase in maximal isometric knee extension torque could be expected in elite judo athletes compared to sea level training. In addition, the rate of torque development, which is more strongly associated with sport-specific performance [31] and more sensitive to detect chronic changes in neuromuscular function after training [32], would increase more after training at moderate altitude than at sea level. It should be noted that there are no controlled studies investigating the effect of judo-specific strength training routines on neuromuscular adaptation, which are typically performed at the end of a special preparation period. Indeed, knowledge of neuromuscular adaptation to judo-specific strength training can help coaches improve their training prescriptions and consequently maximize their athletes’ performance.

## 2. Materials and Methods

### 2.1. Experimental Approach to the Problem

The purpose of a controlled study was to investigate the effects of short-term (i.e., 3 weeks) power-oriented strength training at moderate altitude (2320 m) on neuromuscular variables and strength outcomes. To achieve this objective, elite judokas were randomly assigned to the altitude training group (AL group) and the sea level training group (SL group). The AL group completed power-oriented strength training at the high performance centre Sierra Nevada (Spain) at 2320 m altitude (hypobaric hypoxia). The group SL completed the same training at Spanish Judo Training Centre in Valencia (Spain) at sea level (normobaric normoxia). After power-oriented strength training, a control week was scheduled for both groups. During this week, both groups trained together at Spanish Judo Training Center in Valencia. Pretest and posttest (i.e., one week after descent from altitude) were performed at sea level, regardless of training altitude. A repeated measures design with a within-subjects factor (pretest vs. posttest) and a between-subjects factor was used to determine the main effect of training (pretest and posttest), altitude (sea level vs. altitude), and their interaction on each neuromuscular variable.

### 2.2. Participants

Nineteen elite male judokas (AL group (*n* = 8): 22.1 ± 2.3 years, SL group (*n* = 11): 22.6 ± 4.1 years) participated in the study (Table 1). The participants had at least 10 years of training experience in judo. Their technical level ranged from first to third dan (black belt), and all of them were medalists in the junior or senior division National Championships in Spain, Dominican Republic or Georgia, five of them in the junior or senior division European Cups, five in the Continental Opens, one in Gran Prix, two in the junior division Continental Championships and one in the junior world championship. The competitors were selected if they had no chronic illnesses or recent injuries that could affect their performance, and they were experienced in strength training. Participants had no previous altitude training experience and had not been exposed to altitudes above 1500 m a.s.l. for more than 3–4 consecutive days for at least 2 months prior to the study. The study was conducted at the end of a special preparatory mesocycle in which the main objective was to improve muscular power [5]. They had not performed strenuous exercise for at least two days prior to the pretest. They maintained their usual food intake and abstained from potential ergogenic supplements during the study. Prior to any data collection, all participants were informed of the study protocol and signed a written informed consent form. The study protocol was approved by Institutional Review Board of the University and was in accordance with the Helsinki Declaration.

### 2.3. Experimental Design

During the altitude camp, the judo athletes completed 14 judo (JUDO) and 15 conditioning (CON_TOT_) sessions, consisting of 8 explosive strength-oriented (CON_EX_; Monday, Wednesday, Friday) and 7 metabolic sessions (CON_ME_; Tuesday, Thursday, Saturday). In both groups, the average session duration was 102.55 ± 14.48 min for JUDO and 78.54 ± 4.75 min for CON_TOT_. JUDO sessions were agreed upon by the trainers, maintaining the same exercises and volume. In both altitudes, a general warm-up consisted of 5 min of submaximal running at 7 to 8 km/h, followed by 5 min of dynamic stretching of all major muscle groups. In addition, a specific warm-up consisting of countermovement jumps on the Smith machine was performed before CON_EX_ (2 × 5 repetitions, with load increase from 0 to 20 kg). Perceived exertion was assessed within 30 min of the end of CON_EX_, CON_ME_ and JUDO sessions using a rate of perceived exertion (RPE) scale of 1–10 (RPE-10) to monitor exercise intensity. All participants had prior experience using the RPE-10. The AL and SL groups’ ratings of perceived exertion during the training program are shown in Table 2. There were no significant differences between the SL and AL groups in the rating of perceived exertion (RPE-10) according to each strength training modality (i.e., CON_EX_, CON_ME_). A significant difference was observed only during the JUDO training sessions (*p* = 0.033) due to altitude acclimatization (Table 2).

During CON_EX_, judokas performed jumps with countermovement (CMJ) on the Smith machine (6 repetitions, 4–6 sets, ~35–40% of the repetition maximum (RM), 4 min rest between sets). Five minutes later, participants performed free half squats or deadlifts (2 repetitions, 70–90% RM, 3–4 sets, 4 min rest between sets), immediately followed by ippon seinage at maximal intensity. The training load displaced during all CMJs (∼35–40% 1RM) was estimated from the load associated with 1.2 ms^−1^ mean propulsive velocity from the individual load-velocity relationship [33]. To do this, a linear regression model from a three-load incremental test was fitted each Monday after warm-up and used to estimate the new weekly external load corresponding to a mean barbell propulsive velocity of 1.2 ms^−1^. Table 3 shows the CON_EX_ training load used for the groups AL and SL. Wednesday and Friday training loads were estimated during pre-training, which allowed the participants to improve the velocity as their explosive leg extension capacity increased in both altitudes.

During CON_ME_, judoka performed a high-intensity circuit-training routine of short, medium, or long duration (10–20 min), as well as compensatory exercises. The content of the physical conditioning training was designed and supervised by the research team. In general, 4–12 functional and/or technical exercises performed sequentially (20 min) or intermittently (3–4 blocks).

After the altitude camp, there was a control week during which participants completed the same programme at sea level. During this week, participants completed 5 sessions at JUDO and 5 sessions at CON_TOT_ (3 CON_EX_ and 2 CON_ME_). The average session duration was 115.60 ± 10.81 for JUDO and 76.80 ± 6.34 for CON_TOT_. Similarly, at the end of the control week, there were no differences in RPE between groups according to the strength training modalities or JUDO sessions (Table 2).

### 2.4. Testing Procedures

Neuromuscular tests performed before (pretest) and one week after descent from altitude (posttest) in both conditions consisting of (a) soleus H-reflex assessment, (b) maximal and (c) explosive voluntary isometric knee extension with assessment of interpolated twitch technique, and (d) evoked contractions of the quadriceps femoris muscle (Figure 1). Participants’ habituation to electrical stimulation was performed one day before the start of the experiment. Then, subjects performed two laboratory sessions in a fixed order on two consecutive days to avoid fatigue or post-activation potentiation. The H-reflex was measured in the morning (Figure 1A), and voluntary and evoked contraction measurements were performed in the afternoon (Figure 1B). All neuromuscular measurements were performed under both conditions, i.e., at sea level and at altitude, at approximately the same time of day and by the same investigators.

#### 2.4.1. H-Reflex Measurements

H-reflexes were elicited from the soleus muscle in the supine position after 10 min of rest in the same position. First, the optimal position for percutaneous electrical stimulation of the tibial nerve was determined with single rectangular pulses (1 millisecond (ms)) delivered to the right tibial nerve via a surface cathode (30 × 24 mm; Kendall, Covidien, Mansfield) manually pressed into the popliteal fossa and with the anode (50 × 50 mm; Axelgaard Manufacturing, Co, LTD, Fallbrook, CA, USA) placed at the patella. Electrical activity of the soleus muscle was recorded with pre-amplified, self-adhesive, wireless electrode (dimension: 27 × 37 × 15 mm; mass: 14.7 g; interelectrode distance: 10 mm; baseline EMG noise: 750 nV; common mode rejection ratio: ≤80 dB; Delsys Inc., Boston, MA, USA), placed on one third of the distal length between the medial malleolus and the medial epicondyle of the tibia, following the recommendations of SENIAM [34]. Electromyographic data were recorded with the PowerLab system (16/30-ML880/P, ADInstruments, Bella Vista, Australia) at a sampling frequency of 2000 Hz and filtered with a bandpass 10–500 Hz. EMG signals were recorded and analyzed using LabChart7 software (ADInstruments, Bella Vista, Australia).

To determine the M-wave and H-reflex recruitment curves, the tibial nerve was stimulated with a constant current electrical stimulator (DS7A; Digitimer, Hertfordshire, United Kingdom). The H-reflex recruitment curve was determined by gradually increasing the current intensity (0.1-mA, every 10–15 s) until the maximal response was identified. The M-wave recruitment curve was obtained by gradually increasing the current intensity (5 mA, every 5–10 s) until a plateau of the response was reached. Stimulation intensities to elicit maximal M waves were set at 32.2 ± 7.2 mA for the pretest and 32.1 ± 7.6 mA for the posttest (*p* = 0.945). Stimulation intensities to elicit maximal H waves were set at 11.6 ± 3.5 mA for the pretest and 11.3 ± 4.2 mA for the posttest (*p* = 0.732).

The highest peak-to-peak amplitude for the M-wave (_SO_M_MAX_) and H-reflex (H_MAX_) was determined from the unrectified EMG signals. Based on the soleus EMG response, the following parameters were calculated: (i) the peak-to-peak amplitude of the H wave (H_MAX_), i.e., the amplitude from the positive to the negative peak of the highest H wave, and (ii) the peak-to-peak amplitude of the M wave (_SO_M_MAX_), i.e., the amplitude from the positive to the negative peak of the highest M wave. Then, the ratio between H_MAX_ and _SO_M_MAX_ (H_MAX_ ∙ _SO_M_MAX_^−1^) was calculated and retained for analysis.

#### 2.4.2. Voluntary and Evoked Isometric Knee Extension

During the measurements, which included voluntary and evoked knee extensions (i.e., right leg), the subject sat in an isometric torque measuring device equipped with a strain gauge force sensor (MES, Maribor, Slovenia). Our custom-built torque measuring device was equipped with 2 levers. The two levers were rigidly connected at a fixed angle (60 degrees). The force sensor (MES) was attached to the frame perpendicular to the axis of rotation and at a fixed (non-adjustable) distance from this axis. The second lever was adjustable to the leg length of the subject. Therefore, the lower leg was always attached to the isometric frame at the same position, 1 cm above the ankle joint (i.e., the lateral malleolus). Two lever systems guaranteed the constant moment arm of the force sensor to the moment at the knee (i.e., the moment of a force measured by the sensor). Subjects were seated in an isometric torque-measuring device with the right knee at 60° (0° full extension). During voluntary and evoked knee extension, subjects’ backs were supported and their hips were firmly fixed. In addition, they were secured to the isometric frame at the pelvis and across the chest to ensure that the upper body did not contribute to the knee extension torque.

While siting, the optimal position for percutaneous electrical stimulation of the femoral nerve and the required intensity were determined. Stimulation was performed with single or double square pulses (1 ms) delivered from a constant current stimulator (DS7A; Digitimer, Hertfordshire, UK) to the right femoral nerve via a surface cathode (30 × 24 mm; Kendall, Covidien, Mansfield, TX, USA) manually pressed into the femoral triangle and a 50 × 90 mm anode (Axelgaard Manufacturing Co, LTD, Fallbrook, CA, USA) in the gluteal fold. Electrical activity of the vastus lateralis muscle was measured with a preamplified, self-adhesive, wireless electrode (dimension: 27 × 37 × 15 mm; mass: 14.7 g; interelectrode distance: 10 mm; baseline EMG noise: 750 nV; common mode rejection ratio: ≤80 dB; Delsys Inc, Boston, MA, USA), which was placed according to the recommendations of SENIAM [34]. Electromyographic and mechanical data (i.e., knee extension torque) were recorded using the PowerLab system (16/30-ML880/P, ADInstruments) with a sampling frequency of 2000 Hz. To determine the maximum stimulation intensity when assessing activation levels and evoked contractions, individual stimuli were delivered gradually in 5–10 mA increments until a plateau was reached in the M-wave amplitude of the lateral vastus at rest and the quadriceps twitch. Intensity was then increased by 50% to confirm supramaximal stimulation. The calculated supramaximal intensity was used to electrically trigger the contraction of a relaxed and fully activated muscle (interpolated twitch technique).

Prior to data collection, subjects performed a standardized warm-up consisting of four brief (~5s) submaximal isometric knee extensions at approximately 20%, 40%, 60%, and 80% of maximum, 10–15 s apart. Subsequently, subjects performed 3 maximal isometric contractions (~5s) 1 min apart. Exactly 3 min after the end of the third maximal trial, 2 maximal explosive trials (~3) were performed 1 min apart. Subjects were instructed to exert “as much force as possible, as fast as possible” from a completely relaxed state and were verbally encouraged to ensure maximal explosive effort. Any countermovement before the onset of the explosive contraction was strictly limited.

To determine the percentage of voluntary activation of the quadriceps femoris muscle (%VA), the interpolated twitch technique was used to stimulate the right quadriceps femoris muscle during the second and third knee extensions (Figure 1B). The technique was adapted from the twitch interpolation technique [35]. The supramaximal double-pulse stimulation (two stimuli separated by 10 ms) was delivered during the plateau of voluntary peak knee extension torque (i.e., superimposed Db100) and 3 s after the cessation of contraction (i.e., control Db100). This provided the opportunity to obtain a potentiated mechanical response, reducing the variability in activation level values (VA%). VA% was then calculated as follows VA% = (1 − (superimposed double twitch)/(control double twitch)) × 100, where superimposed double twitch is the amplitude of the twitch elicited by the electrical stimulation on the top of the maximal isometric torque and control double twitch is the amplitude (T_Db100_) of the double twitch delivered to the passive muscle 3 s after the voluntary knee extension.

Mean torque values (T_MVC_; expressed as Nm) and vastus lateralis activity were collected during the second and third trials over the 0.5 s period, immediately before the superimposed double twitch. The raw EMG were filtered (bandpass 10–500 Hz) before calculating root-mean-squared values (RMS) using a 50 ms moving rectangular window (ADInstruments). The obtained RMS value was then normalized to the maximum amplitude of the M-wave (RMS _VL_M_WAVE_^−1^). The highest values from the last 2 maximal trials for each subject were retained for statistical analysis.

Rate of torque values (RTD; expressed as Nms^−1^) were derived from explosive knee extensions. The highest RTD values during explosive knee extension for each subject were selected for analysis. Torque development values represent the average slope of the initial time phase of the torque-time curve between 0 and 75 ms relative to the onset of contraction [36]. The onset of muscle contraction was determined visually/manually by a trained investigator who used a systematic approach and viewed the torque recordings on a consistent scale [37]. Electromyographic data were quantified using the RMS values for the vastus lateralis for a time interval of 0–75 ms and normalized to _VL_M_WAVE_ (RMS_RTD_) for further statistical analysis. Raw EMGs were filtered (bandpass 10–500 Hz) before calculating root-mean-squared values (RMS) using a 50 ms moving rectangular window (ADInstruments).

From the evoked mechanical response elicited by a single supramaximal electrical stimulation of the femoral nerve delivered to the passive muscle exactly 6 s after termination of voluntary contraction, the following parameters were analyzed: (i) the peak twitch torque (T_TW_), i.e., the highest value of the twitch torque curve, and (ii) the twitch contraction time (CT_TW_), i.e., the time from the increase of the initial torque above 5% of the peak T_TW_ to the time point at 95% of the peak T_TW_. From the EMG response elicited by a single supramaximal electrical stimulation of the femoral nerve, the peak-to-peak M-wave amplitude (_VL_M_WAVE_) was calculated as the voltage difference between the two extreme points of the electromyographic curve. The average values calculated from two responses were considered for all parameters obtained.

#### 2.4.3. Statistical Analysis

Data are presented as mean ± standard deviations (SD). Statistical analyzes were performed using the Statistical Package for Social Sciences (SPSS version 27, Chicago, IL, USA). Normality of the data was checked using the Kolmogorov–Smirnov test. VA% data were log transformed but remained non-normally distributed even after log transformation. A repeated measure design was used to assess training (within subject factor) × altitude (between subject factor) interaction. Paired sample T-test was used to test within group changes over time (pretest vs. posttest). Effect size (ES) was calculated using Cohen’s d. The criteria to interpret the magnitude of the Cohen’s d were as follows: <0.19 = trivial, 0.2–0.59 = small, 0.6–1.19 = moderate, 1.2–1.99 = large, and >2 = very large [38]. Percentage differences ((post-test mean − pretest mean) × 100) were also calculated. Pearson correlation coefficients were calculated for pairs of variables. For the non-normally distributed data, Kruskal-Wallis ANOVA and a Wilcoxon signed-rank test were used for between-group and within group differences. The significance level was set at *p* < 0.05. The level of *p* values less than 0.1 was accepted as trend.

## 3. Results

### 3.1. Maximal Isometric Strength of the Knee Extensors

No difference was found between groups in T_MVC_ at baseline (t = 0.865; *p* = 0.40; Table 4). Power-oriented strength training did not increase T_MVC_ in elite judo athletes, neither in the SL group (t = 1.64; *p* = 0.132; Cohen’s d: 0.5; small) nor in the AL group (t = 0.492; *p* = 0.638; Cohen’s d: 0.17; trivial). There were also no statistically significant differences between groups in changes in T_MVC_ as a function of environmental conditions (F_1,17_ = 0.17; *p* = 0.69; Table 4).

### 3.2. Muscle Activity and Voluntary Activation Level

No differences were observed between groups in terms of RMS_MVC_ (t = 1.05, *p* = 0.30; Table 4), RMS_RTD_ (t = −0.87, *p* = 0.40; Table 4), and _VL_M_WAVE_ amplitude (t = −0.34, *p* = 0.72; Table 4) at baseline. Power-oriented strength training performed at the end of a special preparation mesocycle did not increase muscle activity during maximal and explosive muscle contraction in elite judo athletes, neither in the SL group (RMS_MVC_: t = 1.46; *p* = 0.18; Cohen’s d: −0.44; small; RMS_RTD_: t = −0.69; *p* = 0.51; Cohen’s d: −0.21; small) nor in the AL group (RMS_MVC_: t = −1.89; *p* = 0.10; Cohen’s d: −0.66; moderate; RMS_RTD_: t = −0.31; *p* = 0.76; Cohen’s d: −0.11; trivial).

There was no training × altitude interaction for RMS_MVC_ (F_1,17_ = 1.74; *p* = 0.21) or RMS_RTD_ (F_1,17_ = 0.06; *p* = 0.80). Similarly, no difference was found between groups for voluntary activation level (VA%) at baseline (Z = −0.29; *p* = 0.77; Table 4). Power-oriented strength training did not induce significant changes in VA%, neither in the SL group (Z = −0.45; *p* = 0.66), nor in the AL group (Z = -1.54; *p* = 0.66). The _VL_M_WAVE_ did not change after training in either the SL group (t = 0.38; *p* = 0.71; Cohen’s d: 0.11; trivial) or the AL group (t = 0.52; *p* = 0.62; Cohen’s d: 0.18; trivial). Accordingly, no difference was found between groups in the M-wave amplitude of the vastus lateralis by training under different altitude conditions (training × altitude: F_1,17_ = 0.17, *p* = 0.81).

### 3.3. Explosive Isometric Strength of the Knee Extensors

For RTD, no difference was found between groups at baseline (t = 0.44; *p* = 0.66; Table 4). On the contrary, after training camp, the difference between groups of judo athletes in terms of change in RTD as a function of altitude was significant (F_1,17_ = 4.7; *p* = 0.04). Specifically, power-oriented strength training did not produce significant changes in RTD in either the SL group (t = 1.44; *p* = 0.182) or the AL group (t = −1.62; *p* = 0.148), while Cohen’s d showed a positive meaningful effect (Cohen’s d: 0.43, small) in the SL group and a negative meaningful effect (Cohen’s d: −0.58; small) in the AL group.

### 3.4. Spinal Excitability

No differences were found between groups on any variable of soleus H-reflex at baseline (t = −0.78–1.36; *p* = 0.2–0.67; Table 5). Power-oriented strength training had no significant effect on soleus H-reflex adaptation in elite judo athletes, neither in the SL group (t = 0.1–−0.26; *p* = 0.85–0.92) nor in the AL group (t = −0.26–−1.06; *p* = 0.33–0.86). Furthermore, there was also no significant training × altitude interaction on the H_MAX_∙ _SO_M_MAX_^−1^ (F_1,16_ = 0.05; *p* = 0.83; Table 5).

### 3.5. Contractile Properties of the Quadriceps

No between-group differences were found for quadriceps twitch (T_TW_) variables elicited by single supramaximal stimuli at baseline (T_TW_: t = 0.47, *p* = 0.64; CT_TW_: t = 0.02, *p* = 0.98; Table 6). Similarly, no difference was found between groups in double twitch torque (T_Db100_) at baseline (t = 0.12, *p* = 0.91; Table 6). Power-oriented strength training had no significant effect on peak twitch torques, either in the SL group (T_TW_: t = 1.64, *p* = 0.132; Cohen’s d: 0.50, small; T_Db100_: t = 1.81; *p* = 0.10; Cohen’s d: 0.55; small), nor in the AL group (T_TW_: t = 0.22, *p* = 0.83; Cohen’s d: 0.08; trivial; T_Db100_: t = 0.11, *p* = 0.92; Cohen’s d: 0.04; trivial). Furthermore, there was also no significant altitude × training interaction on the T_TW_ (F_1,17_ = 0.64; *p* = 0.43) nor T_Db100_ (F_1,17_ = 0.94; *p* = 0.35). In contrast, power-oriented strength training induced a significant increase in CT_TW_ in the AL group (t = 2.58, *p* = 0.04, Cohen’s d: 0.91; moderate), but not in the SL group (t = −0.36, *p* = 0.73, Cohen’s d: −0.11; trivial). The between-group difference in changes in CT_TW_ as a function of altitude was on the edge of significance (training × altitude: F_1,17_ = 3.5; *p* = 0.077).

### 3.6. Relationship between Contractile Properties of the Quadriceps and Voluntary Torque Production

In the AL group, changes in T_TW_ (R = 0.78; *p* = 0.022) and TDB100 (R = 0.805; *p* = 0.016; Figure 2 correlated significantly with posttest changes in RTD, whereas no such statistically significant correlation was observed in the SL group.

## 4. Discussion

To our knowledge, this is the first study to examine the neuromuscular adaptations of a three-week power-oriented strength training program at natural altitude. Knee maximal isometric strength did not change significantly after training, at altitude, or at sea level. On the contrary, altitude had a significant and opposite effect on the rate of torque development in the knee extensors (training × altitude; *p* = 0.04; Table 4). In more detail, power-oriented strength training induced a positive meaningful effect (Cohen’s d: 0.43, small) in the SL group and a negative meaningful effect (Cohen’s d: −0.58; small) in the AL group. Moreover, after explosive and endurance strength training at moderate altitude, CT_TW_ increased significantly only in the AL group (~8%, *p* = 0.036; Cohen’s d: 0.91; moderate), whereas in the SL group, twitch contraction time remained almost unchanged (Table 6). Consequently, in the AL group, the decrease in knee extensor torque development (RTD) rate paralleled the decrease in peak twitch and double twitch torques (Figure 2). It appears that power-oriented strength training at altitude produces a greater change in contractile muscle properties, probably due to greater peripheral fatigue, than similar training at sea level. These results suggest that power-oriented strength training performed at moderate altitude has a more intense effect on contractile muscle properties than similar training performed at sea level. Although a potentiated effect of power-oriented training at altitude on explosive movements, i.e., jump height and peak velocity in the countermovement jump, has already been observed [12], this cannot be explained by the changes in neuromuscular variables obtained in our study. In particular, it seems that the results obtained by Almeida et al. [12] immediately after altitude training favored explosive strength adaptation due to the hypoxic environment, whereas the neuromuscular adaptation obtained one week later did not. It should be noted that the magnitude of power output during the countermovement jump is influenced not only by passive stiffness, fiber type composition, cross-bridge kinetics, and neural drive during the concentric phase, but also by proper muscle stiffness regulation during the eccentric phase of the jump [39]. Furthermore, the mechanical power output differs at different knee angles due to differences in the relative involvement of each muscle and differences in their moment arm relative to the joint [40]. Indeed, the knee angle adopted during the isometric test was 60 degrees, whereas the knee angle changed dynamically during the eccentric and concentric phases of the jump, leading to modifications in the relative contribution of neural and muscular factors to power output. Furthermore, the data presented here include a one-week tapering period, implying that adaptive gain could be affected by hypoxia due to different levels of fatigue induced by training and due to the different training status of elite judo athletes. Therefore, we might suggest that the individual timing of neuromuscular adaptation might mask the obtained results, as elite judo athletes might be at different stages of their adaptation.

Knee isometric strength remained virtually the same after a three-week conditioning training program emphasizing strength training under all environmental conditions (changes varied from ~3.5% to ~4.5% for the AL and SL groups, respectively, Table 4). This is similar to previous results observed after short-term (~4-week) strength training in normoxia [1,41], hypoxia [42] and terrestrial altitude [43]. At this point, it should be emphasized that many studies have reported strength increases of 1 RM following short-term strength training (less than 10 weeks), although isometric strength and twitch torque did not change [41,42]. As summarized by Behm and Sale [44], this could be attributed to the movement specificity of strength training, with significantly greater strength gains reported when subjects were tested with similar contraction types and movement speeds used in strength training (isotonic vs. isometric). The strength training used in our study consists mainly of dynamic resistance exercises performed at maximal velocity, whereas maximal strength was tested during isometric contractions. However, it is possible that strength did not increase in the present study due to a plateau in strength gain, as highly trained judokas performed the experiment [45].

Although maximal knee isometric torque remained virtually the same after training in the SL and AL groups, neural changes, although not significant, were similar and evident in both groups. Indeed, training in the SL group induced a small negative meaningful effect (Cohen’s d: −0.44) on RMS_MVC_, while training in the AL group induced a similar but moderate meaningful effect on the same variable (Cohen’s d: −0.66). Strength protocols are known to increase neural drive of agonist muscle [14], thus strength training at moderate altitude was expected to increase neural drive of agonist muscle due to the previously observed increase in spinal excitability [23,24], the increase in metabolic load [46], and the earlier recruitment of high-threshold motor units [28]. Similar to our study, a decrease in EMG during maximal isometric contraction under normoxia was observed in the Thorstensson et al. [47] study following a similar but longer (eight-week) progressive strength training protocol with a concomitant increase in torque production. The authors attributed this phenomenon to an increase in intra-muscular coordination due to an increase in desynchronization of motor units. Indeed, it has already been shown by Behm and Sale [44] that force production is greater and more consistent with asynchronous versus synchronous electrical stimulation. Therefore, we could speculate that a three-week power-oriented strength training improves intramuscular coordination in elite judo athletes, as isometric strength remained the same regardless of training altitude, despite the reported small and moderately negative meaningful effect on posttest EMG amplitude. An additional argument for the improved intramuscular coordination after training is also the unchanged voluntary activation level, which reached pre-training values in both groups of elite judo athletes (Table 4). Additionally, the highly trained judo athletes in our study may not have been able to quantitatively increase their neural drive during maximal isometric knee extension, as their voluntary activation levels were above 90% [41]. Nevertheless, our results suggest that moderate altitude may not act as a main trigger for differential neural adaptations after explosive and endurance strength training in elite judo athletes. Moreover, the soleus H reflex remains almost unchanged after training in both the AL and SL groups. Our results agree with Colomer-Poveda et al. [17], who showed that four weeks of strength training under hypoxia induces an increase in maximal isometric contraction of the soleus without concomitant changes in the H-reflex.

On the other hand, the different environmental conditions seem to have a significant and opposite effect on the rate of torque development in the knee extensors (*p* = 0.04; Table 4). Indeed, training in the SL group induced a small but positive meaningful effect (Cohen’s d: 0.43) on RTD, while training in the moderate altitude induced a small but negative meaningful effect on the same variable (Cohen’s d: −0.58). Similarly, it was previously shown that RTD decreased more after training in hypoxia compared to normoxia, while maximal torque production was not differentially affected [48]. RTD is influenced by passive stiffness, fiber type composition, cross-bridge kinetics, and neural drive changes [49,50]. The duration of our training intervention (three weeks) may not be sufficient to observe changes in passive stiffness and fibre type composition, thus the RTD changes in the present study may be attributed to changes in neural drive and muscle contractile characteristics. The latter is the most obvious candidate after training at moderate altitude, as the decrease (change) in RTD parallels the (change in) peak twitch torques, i.e., T_TW_ and T_DB100_ (Figure 2) in the AL group. Indeed, elite judoka in whom training at altitude caused a greater change in contractile muscle properties had a greater loss of rapid torque development than those with a small change in contractile muscle properties. This could be related to the fact that neuromuscular adaptation, i.e., fatigue, was influenced not only by environmental conditions and training background, but also by the training level of the participant. Highly trained participants (stronger participants) showed a stronger loss of RTD compared to weaker athletes, since moderated negative correlation at the edge of significance was found between the absolute maximal isometric knee extension torque and the relative decrease in RTD after training at altitude (R = −0.589; *p* = 0.098). Indeed, the relative strength of the weakest judo athletes in the AL group was 3.18 Nm/kg, while the strongest reached 4.45 Nm/kg (within-group range was 1.27 Nm/kg). This notion can be further supported by the study that showed that after a fatiguing exercise (three sets of 31 isokinetic concentric knee extensions at 60° s^−1^) explosively trained main subjects showed a decrease in twitch torque and maximum rate of twitch development, while no change was observed in endurance trained athletes [51].

Moreover, in the AL training group, twitch contraction time increased significantly after training (Table 6), suggesting an alteration in Ca^2+^ kinetics. Indeed, chronic hypoxia was previously shown to increase the production of reactive oxygen species and Pi due to higher intensities of muscle contractions, which consequently slows down the sarcoplasmic reticulum Ca^2+^ release [52]. Therefore, the change in Ca^2+^ kinetics could be attributed to low frequency peripheral fatigue–LFF, which has previously been observed following strength exercise in hypoxia [53,54]. Low-frequency fatigue is characterized by a relative loss of strength at low stimulus frequencies and slow recovery over the course of hours or even days [55]. It appears that our resistance training protocol performed at altitude induced greater LFF in some, but likely not all, elite judo athletes (Figure 2), which was still detectable in posttest RTD (Figure 2).

In contrast, similar training at sea level did not result in a significant change in twitch time parameter and no correlation was observed between RTD and twitch torques. Thus, the present data suggest that power-oriented strength training at altitude may result in specific alterations in contractile properties in some participants, likely due to greater low-frequency peripheral fatigue [56]. Indeed, Márquez et al. [54] have previously shown that a single high-intensity circuit training session (e.g., strength endurance), which induces greater metabolic responses and higher RPE than traditional strength training, causes greater peripheral fatigue than a traditional strength training session.

There are some limitations to our study that should be addressed. First, neural and muscular adaptation was assessed during isometric contractions, whereas dynamic muscle contraction training was performed at higher contraction velocities. Second, the relatively short training period (three weeks) was probably insufficient to achieve complete neuromuscular adaptation. Third, reduced oxygen levels during training at moderate altitude were not closely monitored because training was conducted at terrestrial altitude. Fourth, any observed superior adaptation cannot be attributed exclusively to training in hypoxia, as elite judo athletes were also chronically exposed to a hypoxic environment at rest. Fifth, is the time interval selected for the rate of torque development. We selected only the first 75 ms of the increasing torque-time curve, although the relative contribution of central and peripheral factors by the increasing torque-time curve is different [57]. Moreover, a final limitation is the fact that the potential influence of environmental hypoxia on acute and chronic responses to strength training and on post-altitude muscle behavior remains largely unexplored [12]. Accordingly, a dose–response relationship has not yet been established. This raises the important question of whether there is an optimal acute exposure to hypoxic stimuli that induces the greatest increase in performance during explosive muscle actions.

## 5. Conclusions

In conclusion, altitude power-oriented strength training did not induce substantial changes in neuromuscular parameters. A minor, probably individual, peripheral adaptation in elite judo athletes altered the rate of torque development during explosive isometric knee extension after altitude training. The individual response could be attributed to differences in the training status of elite judo athletes, as a greater decrease in twitch peak torque was observed in some, but not all, elite judo athletes. According to the results of our study, we suggest that during altitude training, the individual neuromuscular response to power-based training should be carefully monitored to avoid major changes in twitch contractile properties (i.e., increase in twitch contraction time). Therefore, neuromuscular assessment should be performed at different time points during and after altitude training to find an optimal relationship between the duration of hypoxic exposure and neuromuscular adaptation.

## Figures and Tables

**Figure 1 ijerph-18-06777-f001:**
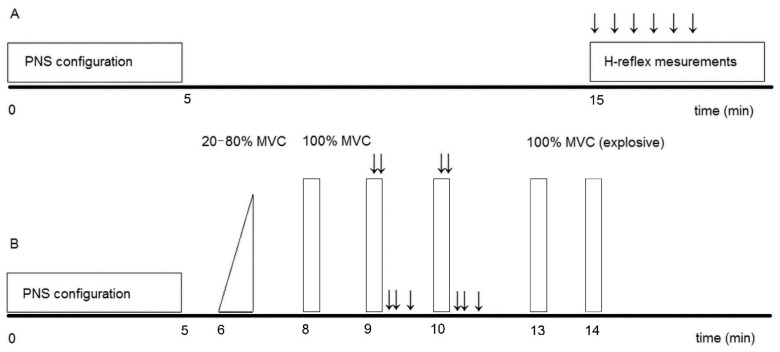
Overview of the neuromuscular testing protocol (panel (**A**)-H-reflex, panel (**B**)–maximal and explosive isometric knee extensions). Black arrows represent peripheral (tibial (A) and femoral (B) nerve stimulations; ↓↓, double supramaximal electrical stimulation (10 ms interspersed interval); ↓, single supramaximal electrical stimulation; MVC, maximal voluntary knee extension; PNS, peripheral nerve stimulation.

**Figure 2 ijerph-18-06777-f002:**
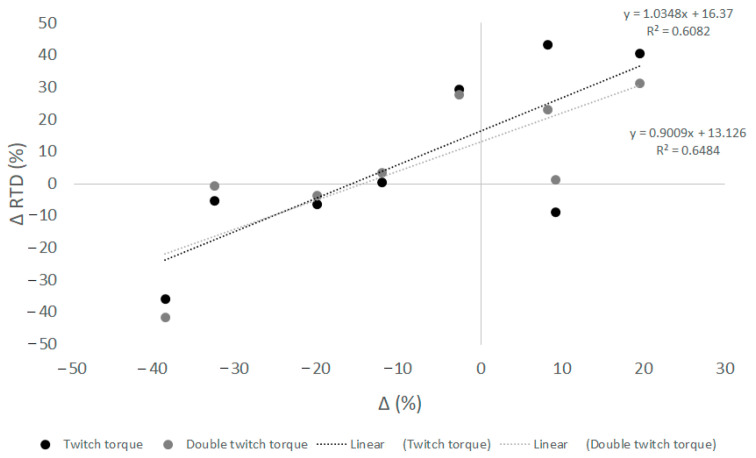
Correlations between relative change in rate of torque development (RTD) of knee extensors and relative changes in T_TW_ (grey circles) and T_DB100_ (black circles) after explosive and endurance training at moderate altitude.

**Table 1 ijerph-18-06777-t001:** General physical characteristics of elite judo athletes pre- and post-strength training in both environmental conditions.

	SL Group (*n* = 11)		AL Group (*n* = 8)	
	Pre	Post	Pre	Post
Height (cm)	176.8 ± 10.3	176.8 ± 10.3	180.7 ± 11.2	180.7 ± 11.2
Body mass (kg)	80.8 ± 19.6	80.9 ± 20.1	85.2 ± 24.0	85.9 ± 22.6
Fat mass (%)	11.7 ± 3.5	11.6 ± 3.0	12.0 ± 4.2	12.0 ± 4.0

SL—sea level, AL—moderate altitude (2320 m a.s.l.).

**Table 2 ijerph-18-06777-t002:** Rating of perceived exertion (RPE-10) in the training and control weeks in both groups.

		SL Group	AL Group	*p* [95% CI]	Cohen’s d
Training	CON_EX_	5.51 ± 0.89	5.70 ± 0.49	0.58 [−0.89, 0.51]	0.27
	CON_ME_	6.65 ± 0.48	6.95 ± 0.63	0.24 [−0.82, 0.22]	0.54
	CON_TOT_	6.08 ± 0.63	6.31 ± 0.46	0.37 [−0.76, 0.30]	0.42
	JUDO	6.78 ± 0.49	6.29 ± 0.44	0.03 [0.05, 0.93]	−1.05
A control week	CON_TOT_	6.40 ± 0.83	6.10 ± 0.84	0.47 [−0.57, 1.17]	−0.36
	JUDO	7.10 ± 0.40	7.27 ± 0.78	0.57 [−0.76, 0.44]	0.28

SL group—elite judo athletes training at sea level, AL group—elite judo athletes training at altitude (2320 m a.s.l.); CON_EX_—explosive strength training; CON_ME_—metabolic session; CON_TOT_—total conditioning training; JUDO—judo training. *p* [95% CI]: *p*-value with the corresponding 95% confidence interval, Cohen’s d—effect size.

**Table 3 ijerph-18-06777-t003:** Training load linked to 1.2 ms^−1^ of the mean propulsive velocity (MPV) throughout the training period.

Load (kg)	SL Group	AL Group
Wednesdays & Fridays	61.59 ± 13.7	59.94 ± 111.24
Monday, 1st week	62.04 ± 15.7	65.56 ± 12.1 **
Monday, 2nd week	65.45 ± 15.6 *	64.11 ± 9.68 *
Monday, 3rd week	71.11 ± 11.9 **	67.31 ± 8.9 **

Wednesdays & Fridays training loads corresponded to the pre-evaluation. * intragroup difference with respect the Wednesday & Friday training loads (** *p* < 0.01, * *p* < 0.05). SL group-elite judo athletes training at sea level, AL group-elite judo athletes training at altitude (2320 m above sea level).

**Table 4 ijerph-18-06777-t004:** Isometric knee strength with muscle activity before and after explosive and endurance strength training in elite judo athletes.

	SL Group			AL Group			Training × Altitude
	Pretest [95% CI]	Posttest [95% CI]	Cohen’s d	Pretest [95% CI]	Posttest [95% CI]	Cohen’s d	
T_MVC_ (Nm)	294.6 ± 62.6 [252.5, 336.6]	307.1 ± 68.7 [261.0, 353.2]	0.50	320.2 ± 65.6 [265.4, 375.1]	326.8 ± 58.3 [278.8, 375.6]	0.17	N.S.
RTD (Nms^−1^)	1981.5 ± 562.7 [1604, 2360]	2175.4 ± 362.9 [1932, 2419]	0.43	2105.2 ± 655.4 [1557, 2653]	1845.3 ± 511.1 [1418, 2273]	−0.58	*p* = 0.04
RMS_MVC_ (%)	6.20 ± 1.31 [5.32, 7.08]	5.38 ± 1.64 [4.28, 6.48]	−0.44	7.24 ± 2.92 [4.79, 9.68]	4.68 ± 1.53 [3.40, 5.97]	−0.66	N.S.
RMS_RTD_ (%)	6.52 ± 2.78 [4.65, 8.39]	5.86 ± 1.74 [4.69, 7.03 ]	−0.21	5.57 ± 1.45 [4.36, 6.79]	5.27 ± 1.53 [4.00, 6.55]	−0.11	N.S.
VA (%)	92.7 ± 9.5 [86.3, 99.1]	92.4 ± 5.5 [88.7, 96.1]	−0.03	94.4 ± 3.5 [91.5, 97.3]	92.1 ± 4.7 [88.2, 96.0]	−0.53	N.S.
_VL_M_WAVE_ (mV)	6.09 ± 2.32 [4.53, 7.66]	6.34 ± 1.98 [5.01, 7.67]	0.11	5.71 ± 2.09 [3.96, 7.46]	6.23 ± 2.36 [4.26, 8.21]	0.18	N.S.

Values are expressed as means ± SD. Training × altitude—*p* values of the combined effects of training and altitude on the dependent variables (N.S.—non-significant). T_MVC_—maximal voluntary isometric knee extensions; RMS_MVC_—the root mean square values of the surface EMG attained during maximal isometric contraction; normalised to maximal M-wave amplitude of vastus lateralis muscle; RTD—the average slope of the torque-time curve between 0–75 ms relative to the onset of the muscle contraction; RMS_RTD_—the root mean square values of the surface EMG attained between 0 and 75ms relative to the contraction onset, normalised to _VL_M_WAVE_ amplitude. SL group–elite judo athletes training at sea level, AL group—elite judo athletes training at altitude (2320 m above sea level).

**Table 5 ijerph-18-06777-t005:** Spinal excitability before and after explosive and endurance strength training in elite judo athletes.

	SL Group			AL Group			Training × Altitude
	Pretest [95% CI]	Posttest [95% CI]	Cohen’s d	Pretest [95% CI]	Posttest [95% CI]	Cohen’s d	
H_MAX_ (mV)	1.72 ± 0.80 [1.18, 2.25]	1.75 ± 1.02 [1.07, 2.43]	0.03	1.89 ± 0.84 [1.10, 2.67]	1.83 ± 0.78 [1.11, 2.55]	−0.40	N.S.
_SO_M_MAX_ (mV)	5.32 ± 1.84 [4.10, 6.56]	5.26 ± 1.58 [4.19, 6.32]	−0.06	4.63 ± 1.82 [2.94, 6.31]	4.56 ± 2.07 [2.65, 6.48]	−0.07	N.S.
H_MAX_∙_SO_M_MAX_^−1^	0.33 ± 0.14 [0.23, 0.42]	0.33 ± 0.17 [0.21, 0.44]	0.03	0.41 ± 0.11 [0.31, 0.52]	0.41 ± 0.07 [0.34, 0.47]	−0.09	N.S.

Values are expressed as means ± SD. Training × altitude—significance of the *p* values of the combined effects of factors on the dependent variable (N.S.—non-significant). SL group—elite judo athletes training at sea level, AL group—elite judo athletes training at altitude (2320 m above sea level).

**Table 6 ijerph-18-06777-t006:** Contractile properties of the quadriceps before and after explosive and endurance strength training in elite judo athletes.

	SL Group			AL Group			Training × Altitude
	Pretest [95% CI]	Posttest [95% CI]	Cohen’s d	Pretest [95% CI]	Posttest [95% CI]	Cohen’s d	
T_TW_ (Nm)	78.9 ± 18.0 [66.8, 91.0]	87.7 ± 18.2 [75.4, 99.9]	0.50	83.3 ± 22.1 [64.8, 101.7]	84.9 ± 15.3 [72.1, 97.6]	0.08	N.S.
CT_TW_ (ms)	53.8 ± 6.2 [49.6, 57.9]	53.1 ± 5.6 [49.3, 56.7]	−0.11	53.8 ± 4.4 [50.2, 57.4]	58.1 ± 6.6 [52.6, 63.6]	0.91	*p* = 0.077
T_DB100_ (Nm)	130.1 ± 24.1 [113.9, 146.3]	144.4 ± 32.1 [122.8, 165.8]	0.55	131.6 ± 31.7 [105.1, 158.04]	132.8 ± 31.7 [113.6, 152.0]	0.11	N.S.

Values are expressed as means ± SD. Altitude × training—*p* values of the combined effects of training and altitude on the dependent variables (N.S.—non-significant). T_TW_—twitch torque; CT_TW_—twitch torque contraction time, T_Db100_—double twitch torque. SL group—elite judo athletes training at sea level, AL group—elite judo athletes training at altitude (2320 m above sea level).

## Data Availability

The raw data supporting the conclusions of this article will be made available by the authors, without undue reservation.

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
