# Peer review of "Neuromuscular Adaptations after an Altitude Training Camp in Elite Judo Athletes"

_ijerph, 2021, doi:10.3390/ijerph18136777_

Round 1
Reviewer 1 Report
Include in the study limitations the controversy surrounding the difficulty of strength adaptations at altitude (as opposed to endurance where they are proven).
Author Response
Response to Reviewer 1
First of all, we would like to thank you for the valuable comments that significantly improved our manuscript. We have carefully evaluated all your suggestions and corrections and have responded accordingly. Below are our responses to your comments and suggestion. All changes made in the manuscript are highlighted in red.
Point 1: Include in the study limitations the controversy surrounding the difficulty of strength adaptations at altitude (as opposed to endurance where they are proven).
Response 1: Thank you for your supportive evaluation of our manuscript. We agree with your comment, therefore following text were added “And the final one, is the fact that the potential influence of environmental hypoxia on acute and chronic responses to strength training and on post-altitude muscle behavior remains largely unexplored [12]. Accordingly, a dose-response relationship has not yet been established. This raises the important question of whether there is an optimal acute exposure to hypoxic stimuli that induces the greatest increase in performance during explosive muscle actions.” (see lns 590 – 596).

Reviewer 2 Report
Review of the paper entitled:
Neuromuscular adaptations after an altitude training camp in elite judo athletes
The authors present the results of a 3-week training period in judokas either at sea level or at an altitude of 2320 m on various neuromuscular parameters. The training period had no major effects on the parameters that were measured (rate of force development, peak force, H-reflex, EMG etc.). The paper is well documented and overall it is of very good quality although the authors present mainly negative results. I have general and specific comments.
1- The training protocol appears as a mix between an overload period (due to the training frequency during the weeks) and a power-oriented training (due to the nature of the exercises). This might explain the absence of clear effects. If the goal was to improve power (as a general quality – but it seems that the lower limbs were targeted here), the authors should justify the training frequency during the week. It is also possible that some adaptations mights have been seen “long” after these 3weeks of training if fatigue had accumulated (maybe 1-2 weeks after, like after a tapering period).
2- The training load has been estimated using a force-velocity profile. I would suggest to include an analysis of this curve (Pmax, Vmax and Fmax) through the training period if available because this is something that changed.
3- The results about RFD are not clear. There is a significant interaction (p=0.04) but no effect of training in either group (L405-406; the statistics for the AL group is not presented by the way). Something is wrong here. At least one of the group should show a significant change.
2- It is not clear what the relation between RFD and the twitch parameters means, the number of subjects is quite low to perform a correlation and the r² is only about 0.6, meaning that there are some chances that this result does not hold with more subjects. Is this result discussed?
3- The conclusions section should present the main conclusions, i.e., no changes in the neuromuscular parameters, except maybe in RFD and the force-velocity profile (to be confirmed). For example, is it really important to state that “the relative intensity of strength-based resistance training was matched between environmental conditions.” This is about methodology, this not a conclusion for me.
others comments
L74. Isn’t the recurrent inhibition (Renshaw cells) also involved? And the sarcolemma and axons excitabilities?
L 204. What is meant by compensatory exercises
Table 4. What the values in front of AL represent?
Author Response
Response to Reviewer 2
First of all, we would like to thank you for the valuable comments that significantly improved our manuscript. We have carefully evaluated all your suggestions and corrections and have responded accordingly. Our responses to your comments and suggestions are written in pdf file (Reviewer 2). All changes made in the manuscript are highlighted in red.

Reviewer 3 Report
Lines 139 and 145 different the same idea, it's written totally the same thing. From lines 124 to 129, the situation of the pre-test is not very well understood, as it is carried out before separating the groups or if it is carried out only when the group that was at altitude has descended to sea level.
Author Response
Response to Reviewer 3
First of all, we would like to thank you for the valuable comments that significantly improved our manuscript. We have carefully evaluated all your suggestions and corrections and have responded accordingly. Below are our responses to your comments and suggestions. All changes made in the manuscript are highlighted in red.
Point 1: Lines 139 and 145 different the same idea, it's written totally the same thing.
Response 1: The duplicated sentences were deleted.
Point 2: From lines 124 to 129, the situation of the pre-test is not very well understood, as it is carried out before separating the groups or if it is carried out only when the group that was at altitude has descended to sea level.
Response 2: Pretest and post test (i.e., one week after the AL group descent from altitude) were performed at sea-level. (see lns 126-127)
